

# A mathematical framework for quantifying physical damage over time from concurrent and consecutive hazards

Alessandro Borre[1,2,*], Daria Ottonelli[2], Eva Trasforini[2], Tatiana Ghizzoni[2], Giacomo Zoppi[3], Giorgio Boni[4], and Silvia De Angeli[5,6,*]

[1]University of Genoa, Department of Informatics, Bioengineering, Robotics and Systems Engineering, Via all'Opera Pia 13 16145 Genova, Italy
[2]CIMA Research Foundation, Via Armando Magliotto 2, 17100, Savona, Italy
[3]University of Turin, Department of Economics and Statistics "Cognetti de Martiis", Lungo Dora Siena, 100A, 10153, Torino, Italy
[4]University of Genoa, Department of Civil, Chemical and Environmental Engineering, Via Montallegro 1, 16145 Genova, Italy
[5]Université de Lorraine, CNRS, LIEC, F-54000 Nancy, France
[6]Université de Lorraine, LOTERR, F-57000 Metz, France
[*]These authors contributed equally to this work.

**Correspondence:** Silvia De Angeli (silvia.de-angeli@univ-lorraine.fr), Alessandro Borre (alessandro.borre@cimafoundation.org)

**Abstract.** Space and time play a crucial role in multi-hazard impact assessment. When two or more natural hazards occur at the same location simultaneously or within a short time frame, the physical integrity of assets and infrastructures can be compromised and the resulting damage can be greater than the one generated by individual hazards occurring in isolation. The current literature highlights the lack of quantitative standardised frameworks for multi-hazard impact assessment. This

5 research presents a generalised mathematical framework for quantitatively assessing multi-hazard physical damage on exposed assets, such as buildings or critical infrastructures, over time. The proposed framework covers both concurrent and consecutive hazards, by modelling: (i) the increased damage resulting from the combined impact of two or more concurrent hazards that overlap in space and time, and (ii) the effects of cumulative damage on asset vulnerability and the recovery dynamics in case of consecutive hazards that overlap in space. The framework is applied to a real-world case study in Puerto Rico, including the

10 concurrent wind and flood impacts generated by the passage of Hurricane Maria, as well as the consecutive impacts caused by the subsequent seismic sequence of 2019-2020. Based on simulations performed on a building portfolio, we found that neglecting residual damages caused by the hurricane when assessing the impacts of the subsequent earthquake would lead to a significant underestimation of the overall damage experienced by the assets. By providing a generalised formalisation to perform quantitative multi-hazard impact assessment, able to account for amplification phenomena and recovery dynamics,

15 the framework can offer scientists and decision-makers a comprehensive and deeper understanding of the impacts caused by compound and consecutive events.



# 1 Introduction

In recent years, increasing attention from the scientific community and international frameworks has been drawn to multi-(hazard)-risk assessment and management (Ward et al., 2022). There has been a growing acknowledgement that natural hazard
events may occur simultaneously, in cascade, or cumulatively over time, and can interplay with societal factors such as exposure and vulnerability, to generate complex multi-risk disaster scenarios (de Ruiter et al., 2020; de Ruiter and van Loon, 2022). These compound events, like heavy rainfall and wind extremes from the same convective storm (e.g., Tilloy et al. (2022)), cascading events like earthquakes triggering tsunamis (e.g., Mimura et al. (2011)), or even consecutive independent events (e.g., NASA (2021)) can generate an impact that is different from that of the individual hazards occurring in isolation (Gill and Malamud,
2014). Moreover, in an increasingly interconnected world, natural hazards' impacts cascade across geographical and sectoral boundaries, leading to considerable challenges for disaster risk managers, including emergency management agencies, asset managers, and operators of critical infrastructures and lifelines (Hochrainer-Stigler et al., 2023; Pasino et al., 2021). Although more complex, approaches that account for multiple hazards and their interconnections better capture the real risk many areas of the world are exposed to and can support the definition of effective disaster risk reduction (Kappes et al., 2012).

In contrast to single-hazard risks, the multi-risk assessment (hazard) poses a series of challenges at each step of the risk or impact assessment, from the hazard modelling to the vulnerability characterisation, until the final risk or impact assessment (Kappes et al., 2012; De Angeli et al., 2022). Although significant improvements have been made in developing approaches that can identify and quantify interrelationships between hazards (Tilloy et al., 2019), as well as understanding their spatial and temporal overlap dynamics (Claassen et al., 2023), the quantification of impacts arising from multiple hazards and the
development of generalised models for assessing multi-hazard physical damage remain a partially unexplored domain (Gentile et al., 2022). When modelling physical impacts caused by multiple concurrent or consecutive hazards, several aspects can influence the magnitude and nature of the final damage, including the order in which the hazards affect the exposed assets, the time window between the hazards (if any), and the recovery rate of the impacted asset, among others (De Angeli et al., 2022).

This paper presents a generalised mathematical framework for quantitatively assessing multi-hazard direct physical damage on exposed assets, such as buildings or critical infrastructures, over time. Current quantitative multi-hazard damage models developed to asses direct physical damages to assets or infrastructures are available only for specific combinations of asset types (e.g., buildings or bridges, that are the most investigated) and hazards. A comprehensive review of these models is reported by Gentile et al. (2022). From this systematic review, it emerges that the majority of quantitative multi (dual)-hazard physical impact models available are seismic-related, i.e. are models where at least one of the two considered hazards is an
earthquake. To overcome this specificity, the framework presented in this manuscript is general enough to be applied to a wide range of natural hazards, including geological, hydrological, and climatological ones, and to be integrated with existing multi-hazard damage models.

The framework aims at formalising and quantifying the full range of aspects underlined by De Angeli et al. (2022) as crucial for multi-hazard damage assessment, offering a comprehensive time-dependent mathematical description of: (i) *concurrent (or*
*compound) impacts*, where the impacts result from two hazard events that either completely or partially overlap in space and





time, with the second event occurring before the response phase of the first one; (ii) *consecutive impacts*, wherein the impacts are generated by spatially overlapping hazards which occur close enough in time that the assets do not have sufficient time to completely recover before the onset of the second hazard; (iii) *independent impacts*, which are generated by consecutive hazards that occur so distantly in time that the assets have adequate time to restore their original conditions before the second hazard occurs. Specifically, in the case of consecutive events, the impacts of an earlier event are likely to change the vulnerability at the time of the next event (de Ruiter and van Loon, 2022). In addition, recovery of vulnerability to pre-disaster conditions after the event ended plays a key role. Societies and systems that recover quickly from disasters become less vulnerable to the next event than societies that follow a slower recovery path (Di Baldassarre et al., 2018). Despite its importance in a deeper understanding of complex risk dynamics, the modelling of the recovery process remains a significant challenge even from a single-hazard perspective (Mohammadi et al., 2024). Most of the damage models currently used do not address the temporal dimension of post-disaster loss and recovery or treat it in a simplistic fashion (Miles and Chang, 2006; Sarker and Lester, 2019). Very little research has been conducted on how recovery proceeds over time (e.g., Cimellaro et al. (2010); Loos et al. (2023)), or on the social and economic factors affecting the recovery process (e.g., Miles et al. (2019); Koliou et al. (2020); Hariri-Ardebili et al. (2022)). Without aiming to solve this complex issue, the generalised mathematical framework presented in this manuscript has been conceptualised to incorporate recovery dynamics into the quantitative assessment of damage over time. This approach provides a comprehensive understanding of post-disaster loss dynamics.

To encourage widespread practical application, the framework has been implemented as modular Python code, allowing users to customise multi-hazard parameters, vulnerability functions, and recovery dynamics, and perform multi-hazard damage assessments.

This paper is organised as follows: first, we present the conceptualisation of our mathematical framework (Sect. 2), starting with a single-hazard setting (Sect. 2.1) and then progressing to multi-hazards (Sect. 2.2). We focus on the quantification of concurrent (Sect. 2.2.1), consecutive (Sect. 2.2.2) and independent (Sect. 2.2.3) impacts. Next, we illustrate the application of this multi-hazard quantitative impact framework in a real-world case study in Puerto Rico, covering the concurrent wind and flood impacts from Hurricane Maria and the consecutive impacts from the 2019-2020 seismic sequence (Sect. 3). Finally, we discuss limitations and future developments (Sect. 4) and present the main conclusions (Sect. 5).

## 2  Mathematical Framework for Damage Estimation

We conceptualise a mathematical framework to quantitatively estimate the impact over time $t$ on an asset or infrastructure $j$ caused by one or more natural hazards. More specifically, we focus on the quantification of the direct physical damage $d_j(t)$, which can be also seen as an indirect measure of the physical integrity $y_j(t)$ of the asset, existing the relationship:

$$y_j(t) = 1 - d_j(t), \quad y_j \in [0, 1] \tag{1}$$

$y_j(t) = 1$ indicates no impact on the element's physical integrity and $y_j(t) = 0$ represents complete destruction. The concept of physical integrity was introduced by Minciardi et al. (2006) as a proxy to quantify the asset's functionality. However, our





framework does not cover the assessment of asset functionality, as discussed in previous studies (e.g. Disse et al. (2020); Miles et al. (2019)), or the evaluation of systemic impacts and system resilience (e.g., Cimellaro et al. (2008); Reed et al. (2009)). Quantifying asset functionality, which refers to the inherent ability of an asset to perform its designated task (Ellingwood et al., 2016), requires evaluating various factors such as social, physical, economic and accessibility aspects. It also requires estimating indirect impacts resulting from the loss of functionality in other elements of the territorial system to which the asset belongs.

Accordingly, $d_j(t)$ measures the relative (or percentage) damage of the asset. The absolute damage $D_j(t)$ can be obtained by multiplying the relative damage $d_j(t)$ times the exposure value of the asset $E_j(t)$, as outlined in Eq. (2).

$$D_j(t) = d_j(t) \cdot E_j(t) \qquad (2)$$

The exposure value has to be considered as variable over time due to various factors unrelated to hazards, such as restructuring, maintenance, or improvement interventions, as well as due to lack of maintenance or change of use. Moreover, when considering the effects of a hazard, exposure may diminish due to the damages incurred by the hazardous events, it can dynamically change during the whole recovery phase, and can even reach a higher level following the completion of the recovery process. Therefore, it is crucial to emphasise that to perform a dynamic damage assessment, the focus must also be on evaluating how exposure can dynamically change over time.

## 2.1 Single-Hazard Scenario Modelling

To facilitate the understanding of some of the key elements of the proposed framework, we start with the analysis of a single-hazard scenario. Figure 1 illustrates the variation in physical integrity over time $y_j(t)$ for a general asset $j$ that is impacted by a single natural hazard event, for example, a flood, an earthquake or a hurricane.

We start observing the asset at $t = t_0$. Physical integrity is initially assumed to be set to one, that is, $y_j(t_0) = 1$. At a specific time $t_S$, the asset starts to be impacted by the hazard, resulting in a loss of physical integrity $\Delta y_j$ dependent on the intensity of the hazard event, as well as the vulnerability of the asset. The hazard event ends at $t_F$, with a total duration $\Delta t_{EV} = t_F - t_S$. For simplicity and to facilitate the graphical representation, it is assumed that the loss of physical integrity occurs integrally at the beginning of the event. However, in many real cases the maximum loss is reached only when the hazard reaches its peak (that is, the maximum of its intensity) and sometimes, as in the case of earthquakes, physical damage can manifest even after the end of the event (e.g., the collapse of buildings hours or days after the main shock) (Raghunandan et al., 2015). The temporal evolution of $y_j(t)$ during the event phase could be relevant only in the case of concurrent events that have a non-null interarrival shorter than $\Delta t_{EV}$. In the scheme adopted for concurrent events (see Fig.3), such interarrival is always considered negligible. Furthermore, the duration of the event is assumed to be comparable to or shorter than the response phase. This implies that we primarily focus on so-called sudden-onset hazards, for which the event phase typically lasts no longer than a few days. In contrast, in the case of events such as drought or subsidence, the event phase can last for years, and the distinction between the event, response, and recovery phases is not always easily distinguished (Terzi et al., 2022).



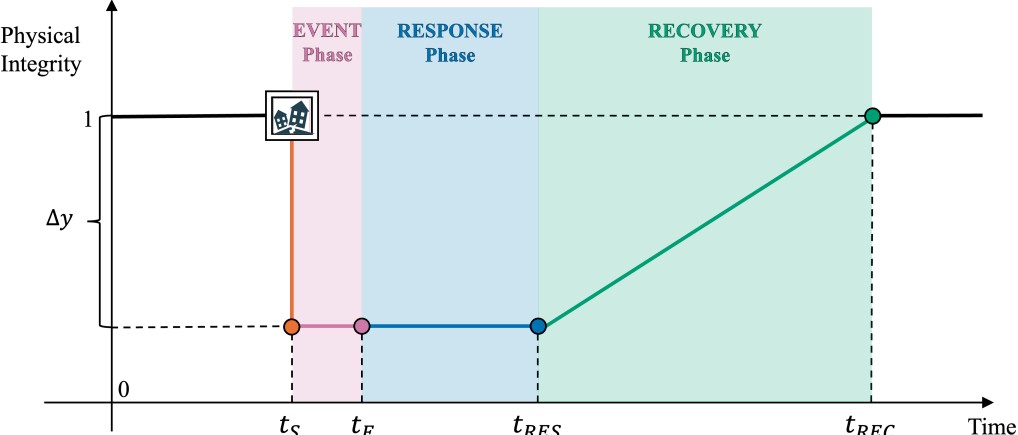

**Figure 1.** Variation of the physical integrity over time for a general asset $j$ which is impacted by a single natural hazard event.

When the event ends, the response phase starts. It consists of actions that are aimed at saving lives, containing losses, and restoring essential services. Response actions may include activating the emergency operations centre, evacuating populations, setting up temporary shelters, providing medical care, and searching and rescuing. The response phase persists until time $t_{RES_1}$ and is represented in Fig.1 by a horizontal blue line. For reasons of simplification, it is assumed that the value of physical integrity $y_j(t)$ remains constant throughout the response phase. Although it is conceivable that in real-world scenarios slight fluctuations might occur, these variations can be deemed negligible compared to the significant changes observed once the asset begins reconstruction during the subsequent recovery phase.

Following the response, the recovery phase begins with the aim of restoring the pre-event level of physical integrity. It is widely acknowledged in the scientific community that the recovery phase begins after the threat to human life has subsided. Nevertheless, the boundaries between the response and recovery phases are blurred, and making a clear distinction is challenging in the context of real disaster development (Mohammadi et al., 2024). The goal of the recovery phase is to bring the affected area back to some degree of normalcy, including restoring basic services and repairing physical, social, and economic damages. During the recovery phase, the reconstruction activities for the asset $j$ begin and continue until time $t_{REC}$. Estimating the duration and dynamics of recovery processes is critical. In Fig. 1, the recovery pattern is represented by a tilted green line. According to Cimellaro et al. (2010), this assumption is valid in contexts where the organisational and response capacity of the system is unknown. However, the recovery function of an asset, $R_j(t)$, can follow various patterns, such as exponential, trigonometric, or linear, with different rates depending on technological, logistic, and economic factors, among others. A trigonometric curve is more suitable for contexts where the initial recovery phase is hampered by a lack of materials and resources needed for reconstruction. If appropriate political measures are swiftly implemented and updated civil protection plans are in place, the response phase will be shorter. This will lead to a faster start to the long-term recovery phase, which



will initially progress rapidly before slowing down as conditions improve. This scenario is best represented by an exponential recovery function Cimellaro et al. (2010).

**Table 1.** Key concepts and their definitions as used in this paper

| Element | Description |
|---------|-------------|
| Physical integrity $y_j(t)$ *(Fig.1)* | Complementary to the relative damage $d_j(t)$, it measures how much an asset remains integer in its physical components. It can vary continuously from 0 to 1, with 1 indicating no impact on the element's physical integrity and 0 representing destruction. |
| Event Phase $\Delta t_{EV}$ *(Fig.1)* | Time window between the beginning and the end of a hazardous event. For sudden events, such as earthquakes or flash floods, it can span from seconds to minutes. For long onset events, such as riverine floods or droughts, it can span from days to years. |
| Response Phase $\Delta t_{RES}$ *(Fig.1)* | Phase that immediately follows the conclusion of the event. It consists of actions that are aimed at saving lives, containing losses, and restoring essential services. Response actions may include activating the emergency operations centre, evacuating populations, setting up temporary shelters, providing medical care, and searching and rescuing. |
| Recovery Phase $\Delta t_{REC}$ *(Fig.1)* | Phase that follows the response, starting when the threat to human life has subsided. It aims to bring the affected area or asset back to some degree of normalcy, including the restoration of basic services and the repair of physical, social, and economic damages. |
| Concurrent Impacts *(Fig.2, panel a)* | Impacts resulting from two hazard events that either completely or partially overlap in space and time. The second event occurs simultaneously with the first one or during its response phase. The challenge is to estimate the overall amplification of physical damage due to the superimposition of the two loads on the same assets. |
| Consecutive Impacts *(Fig.2, panel b)* | Impacts generated when spatially overlapping hazards occur close enough in time that the assets do not have sufficient time to completely recover before the onset of the second hazard. The main challenge is to estimate residual damage according to the recovery dynamic of the asset and understand how this residual damage can influence the vulnerability to the second hazard. |
| Independent Impacts *(Fig.2, panel c)* | The two impacts are independent of each other since they were generated by hazards that occurred so distantly in time that the assets had adequate time to restore their physical integrity before the second hazard occurred. |

Moreover, at the end of the reconstruction, the assets may not be restored to the same conditions as before the event. Indeed, the goal is usually to rebuild them in a way that enhances their resilience and reduces vulnerability, as emphasised by the Build Back Better theory (see, e.g., Kennedy et al. (2008); Saya et al. (2017); Hallegatte et al. (2018)). This improvement can be

reflected in a change in both the exposure, in terms of the total economic value of the assets, and the asset's vulnerability.

To summarise, given a generic hazard event able to generate a loss of physical integrity on a generic asset $j$, the following three main phases can be identified:





- the *event phase* $\Delta t_{EV} = t_F - t_S$

- the *response phase* $\Delta t_{RES} = t_{RES} - t_F$

- the *recovery phase* $\Delta t_{REC} = t_{REC} - t_{RES}$

A concise definition of each of these three phases, as well as of physical integrity, is provided in Table 1.

## 2.2 Multi-Hazard Scenario Modelling

Moving from a single to a multi-hazard scenario and building upon the impact mechanisms classification suggested by De Angeli et al. (2022), the framework enables the analysis of three distinct types of impact interactions, as given in Fig. 2:
(i) concurrent impacts, generated by hazards which either completely or partially overlap in space and time; (ii) consecutive impacts, generated when spatially overlapping hazards occur close enough in time that the assets do not have sufficient time to completely recover before the onset of the second hazard; (iii) independent impacts, generated by hazards that occur so distantly in time that the assets have adequate time to restore their physical integrity before the second hazard occurs.

Figure 2 is a flowchart in which ovals represent the initial and final states, and diamonds represent conditions that require a
Boolean decision (that is, 'Yes' or 'No'). The process begins with a specific temporal evolution scenario involving two hazard events, labelled 1 and 2, affecting one or more assets of interest (depicted in the grey oval at the top of Fig. 2). Depending on whether certain temporal overlap conditions are verified, the three different cases of impact interactions shown in the ovals can be identified. We will now detail the two Boolean decision points (diamonds) and discuss the results that result from confirming ('Yes') or rejecting ('No') the statements associated with each of these decision points. The two Boolean decision points are
labelled A and B.





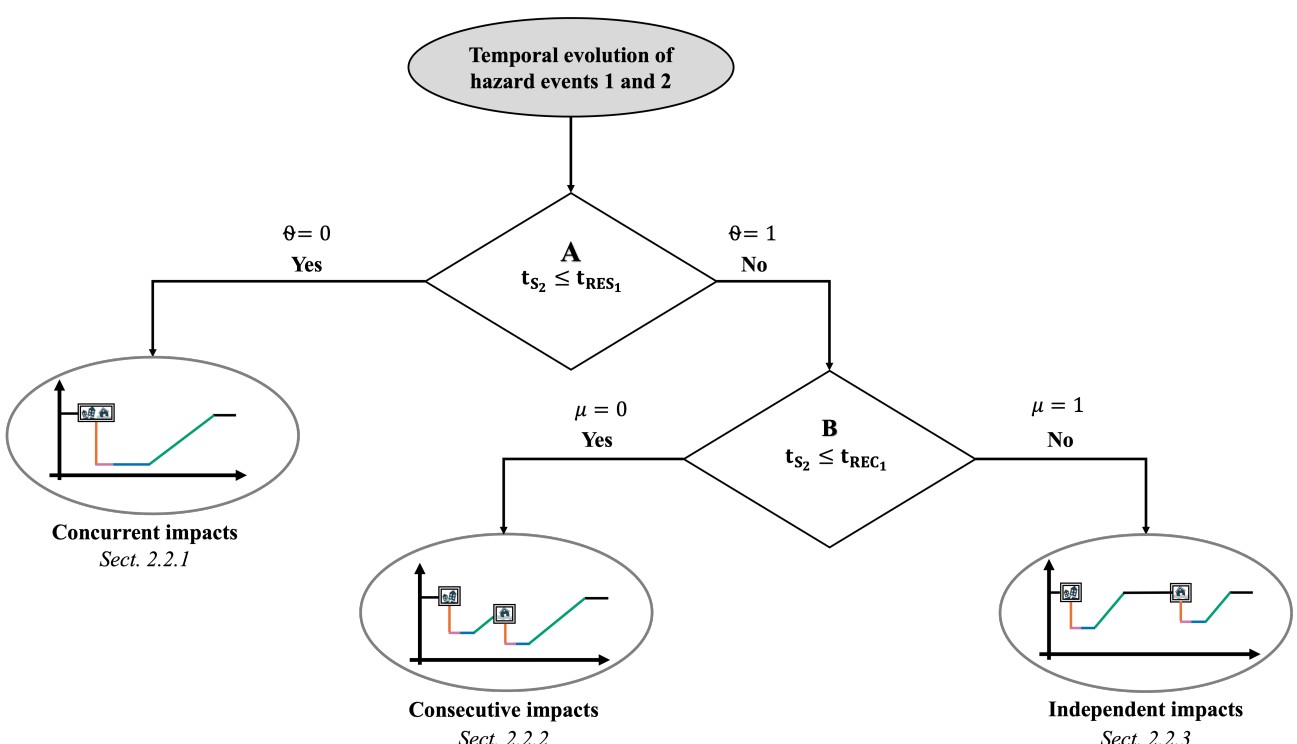

**Figure 2.** Identification of multi-hazard impact interaction typologies resulting from the temporal evolution of two different hazard events. Diamonds A ($t_{S_2} \leq t_{RES_1}$) and B ($t_{S2} < t_{REC1}$) represent Boolean decisions ("Yes" or "No").

The first condition labelled A checks whether the hazard event 2 begins before the end of the event response phase 1', that is, if $t_{S_2} \leq t_{RES_1}$. This condition is mathematically represented by the Heaviside step function $\mathcal{H}$, as shown in Eq. (3).

$$\Theta(t_{S_2}, t_{RES_1}) = \mathcal{H}(t_{S_2} - t_{RES_1}) \tag{3}$$

If A = "Yes", i.e., $\Theta = 0$, hazard event 2 begins before the end of event 1's response phase, and the impacts are considered 165 *concurrent*. In this scenario, a series of impacts simultaneously affect the considered asset, regardless of the relationship between the hazards that generated them. This first type of impact interaction is thoroughly described in Sect. 2.2.1 and visually represented by Fig. 3, panel (a).

If A = "No", i.e., $\Theta = 1$, hazard event 2 begins after the end of event 1's response phase. In this case, two different scenarios can occur depending on the outcome of the second Boolean decision point, labelled B.

The second condition B additionally checks whether hazard event 2 begins before the end of event 1's recovery phase, i.e., if $t_{S_2} \leq t_{REC_1}$. This condition is mathematically represented again by the Heaviside step functions, as shown in Eq. (4).

$$\mu(\Theta, t_{s_2}, t_{REC_1}) = \mathcal{H}(t_{S_2} - t_{REC_1}) \quad if \quad \Theta = 1 \tag{4}$$





If A = "No", i.e., $\Theta = 1$, and B = "Yes", i.e., $\mu = 0$, hazard event 2 begins after the end of event 1's response phase but before the end of event 1's recovery phase. In this case, the impacts are considered *consecutive*. In this scenario, the asset has not had

sufficient time to complete the recovery process from the first event, and when the second event occurs, the asset still possesses a level of integrity lower than one, rendering it more vulnerable to the impacts of the subsequent event. This second type of impact interaction is investigated in detail in Sect. 2.2.2 and visually represented by Fig. 3, panel (b).

If A = "No", i.e., $\Theta = 1$, and B = "No", i.e., $\mu = 1$, hazard event 2 begins after the end of event 1's recovery phase and the impacts are considered *independent*. This last type of impact interaction refers to the fortunate scenario in which events occur

so far apart in time or the recovery has been so fast that, by the time the second event starts, the asset has had sufficient time to fully recover from the previous damage. In this context, the second impact causes damage when the physical integrity of the asset is fully restored from the first event in temporal order, and its current value is again equal to $y_j(t_0)$. The scenario is detailed in Sect. 2.2.3 and visually represented by Fig. 3, panel (c).





**Figure 3.** Variation of physical integrity over time for the different interactions between impacts considered by the general framework. **Panel a**: The impacts are concurrent. The second event in temporal order affects the asset during the response phase of the previous one. The second impact and the preceding one are considered simultaneous. The loss of physical integrity is the amplified combination of the two stressors. Details in Sect. 2.2.1. **Panel b**: The impacts are consecutive. The second event in temporal order affects the asset during the long-term recovery phase of the previous one. Details in Sect. 2.2.2. **Panel c**: The impacts are independent of each other. The second event in temporal order affects the asset when it has already completed the recovery phase of the previous one. Details in Sect. 2.2.3.

### 2.2.1 Concurrent Impacts

Concurrent impacts result from two hazard events that overlap in space and time either completely or partially. The second event occurs simultaneously with the first or during its response phase. Real-world examples include the 2018 events in Indonesia,



where a 7.5 magnitude earthquake in Sulawesi triggered a tsunami, causing extensive coastal damage (Sabah and Sil, 2023; Naik et al., 2023; Opabola et al., 2023). Another example, which does not involve causal relationships between events, is the 2013 earthquake in the Philippines followed by a typhoon before recovery from the earthquake had begun (Lagmay and Eco, 2014; Pormon et al., 2023; Lagmay et al., 2015). In such scenarios, the impacts of both events combine and affect the asset. The primary challenge is to quantitatively assess the overall amplification of physical damage caused by the superimposition of multiple loads.

Direct physical damage over time $t$ to an asset or infrastructure $j$ caused by concurrent impacts is calculated according to Eq. (5).

$$d_j(t)[t_{S_1}, t_{REC_{1,2}}] = \begin{cases} f_1 & \text{if } t_{S_1} \le t < t_{RES_{1,2}} \\ f_1 \cdot R_{j,1,2}(t) & \text{if } t_{RES_{1,2}} \le t < t_{REC_{1,2}} \end{cases} \tag{5}$$

with

$$f_1 = f_{v_{j,1,2}}(h_{1,\max}, h_{2,\max}) \tag{6}$$

According to Eq. (5), at any time $t$ between the start of the first hazard event $t_{S_1}$ and the end of the response phase of the multi-hazard event $t_{RES_{1,2}}$, asset $j$ experiences a level of damage equal to $f_1$. Similarly to the single-hazard scenario (Sect. 2.1), for simplicity and to facilitate the graphical representation, it is assumed that the damage (and therefore the loss of physical integrity) occurs entirely at the onset of the first event. After the response phase of the multi-hazard event concludes (i.e., for $t \ge t_{\text{RES}_{1,2}}$), the asset begins to recover from the concurrent impacts according to the recovery function $R_{j,1,2}(t)$, until $t = t_{REC_{1,2}}$. This evolution of damage in time, as described in Eq. (5), is graphically represented in Fig. 3, panel (a).

Equation (6) represents the damage model used to quantify the effects of two concurrent hazards, referred to as the "concurrent damage model." This model assesses the combined impact of both hazards on asset performance. In the literature, concurrent damages are typically evaluated using bivariate vulnerability or fragility functions. Therefore, we express this model as $f_{v_{j,1,2}}$, where the resulting damage depends on the maximum intensity of each hazard event, $h_{1,\max}$ and $h_{2,\max}$. Gentile et al. (2022) classifies this approach as a "vector-valued fragility model." Bivariate vulnerability functions have been applied in single-hazard scenarios, such as assessing flood damage considering both flow velocity and duration (Elmer et al., 2010; Nofal and van de Lindt, 2020; Ming et al., 2015), and in multi-hazard contexts, such as hurricanes, where structural impacts are evaluated by combining the effects of coastal surges and wind forces (Do et al., 2020; Nofal et al., 2021).

However, there are instances where concurrent damages are calculated using "state-dependent fragility models" (Gentile et al., 2022), where the fragility of an asset to a secondary hazard is conditional on its damage state after the primary hazard. This approach is applied, for example, to assess the impacts of an earthquake-tsunami combination, where the fragility curve for the post-shock tsunami depends on the earthquake-induced damage (Xu et al., 2021; Gómez Zapata et al., 2023).





### 2.2.2 Consecutive Impacts

Consecutive impacts occur when hazard events that spatially overlap occur close enough in time that assets do not have sufficient time to completely recover before the onset of the second hazard. Real-world examples include the sequence of earthquakes and floods in Nepal in 2015. An earthquake with a magnitude of 8.0 struck Nepal in April 2015, causing significant

structural and non-structural damage. The region was then hit by heavy rains and floods in 2017 (Gautam and Dong, 2018). Recovery and reconstruction documents (He et al., 2018; Liu et al., 2021) show that the floods affected structures and areas still recovering from the previous earthquake. Another example is Hurricane Irma and Maria, which devastated Puerto Rico in 2017, followed by an earthquake in the same area two years later. This case will be discussed in detail in Sect. 3. The main challenge with consecutive impacts is to estimate residual damage based on the asset's recovery dynamics and to understand

how the residual damage influences vulnerability to the subsequent hazard.

Direct physical damage over time $t$ to an asset or infrastructure $j$ caused by consecutive impacts is calculated according to Eq. (7).

$$
d_j(t)[t_{S_1}, t_{REC_2}] = \begin{cases} f_2 & \text{if } t_{S_1} \leq t < t_{\text{RES}_1} \\ f_2 \cdot R_{j,1}(t) & \text{if } t_{\text{RES}_1} \leq t < t_{S_2} \\ f_3 \cdot [1 - f_2 \cdot R_{j,1}(t_{S_2})] + f_2 \cdot R_{j,1}(t_{S_2}) & \text{if } t_{S_2} \leq t < t_{\text{RES}_2} \\ \{f_3 \cdot [1 - f_2 \cdot R_{j,1}(t_{S_2})] + f_2 \cdot R_{j,1}(t_{S_2})\} \cdot R_{j,2}(t) & \text{if } t_{\text{RES}_2} \leq t < t_{\text{REC}_2} \end{cases}
\tag{7}
$$

with

$$f_2 = f_{v_{j,1}}(h_{1,\max}) \tag{8}$$

$$f_3 = f_{v_{j,2|d(t_{S_2})}}(h_{2,\max}) \tag{9}$$

According to Eq. (7), at time $t_{S_1}$, due to the impact of the first hazard event, the asset $j$ experiences a level of damage equal to $f_2$. Equation (8) represents a single hazard damage model (e.g. a depth-damage curve for floods or a fragility curve for earthquakes), as a function of the maximum intensity of the hazard events $h_{1,\max}$. For the single-hazard scenario (Sect. 2.1),

it is assumed that the damage (and therefore the loss of physical integrity) occurs entirely at the onset of the event. After the response phase of the first event is complete (that is, for $t \geq t_{\text{RES}_1}$), the asset begins to recover from the impacts of a single danger according to the recovery function $R_{j,1}(t)$.

At time $t_{S_2}$, the asset is still recovering from the previous hazard event when it is impacted by a second event. The damage caused by this consecutive event is expressed as $f_3 \cdot [1 - f_2 \cdot R_{j,1}(t_{S_2})] + f_2 \cdot R_{j,1}(t_{S_2})$. This expression accounts for the sum of

the residual damage at time $t_{S_2}$, denoted by $R_{j,1}(t_{S_2})$, and the additional damage from the second event, which is calculated as the percentage damage caused by the second event, represented by $f_3$, applied to the remaining undamaged portion of the asset, i.e., $[1 - f_2 \cdot R_{j,1}(t_{S_2})]$. This approach prevents the double-counting of damage.





After the response phase of the second event concludes (i.e., for $t \geq t_{\text{RES}_2}$) the asset begins to recover from the consecutive impacts according to the recovery function $R_{j,2}(t)$, until $t = t_{REC_2}$. This evolution of damage in time, as described in Eq. (7),

is graphically represented in Fig. 3, panel (b).

Equation (9) represents the damage model used to quantify the effects of two consecutive impacts, known as the "consecutive damage model." This model is based on a "state-dependent fragility model" (Gentile et al., 2022), denoted as $f_{v_{j,2|d(t_{S_2})}}$. It defines the fragility or vulnerability of an asset to a secondary hazard, conditional on the asset's residual damage state due to incomplete recovery. The resulting damage is a function of the maximum intensity of the second hazard $h_{2,\max}$ and the

residual damage to the asset calculated at $t_{S_2}$ according to the recovery function of the asset. State-dependent fragility models are relatively scarce in the literature and are mainly developed within the seismic field to assess the effects of a primary event followed by aftershocks on an already weakened asset (Li et al., 2014; Aljawhari et al., 2021). Another application in the seismic domain involves evaluating the worsening condition of assets, often bridges, due to factors like time and corrosion. In this scenario, the primary seismic event impacts an asset in a deteriorated state, resulting in increased damage (Otárola et al.,

255    2022).

### 2.2.3 Independent Impacts

In the case of consecutive hazards where the second hazard event occurs after the end of recovery of the first one, the impacts can be considered independent. In such a case, the evolution of the damage over time can be seen as a series of single-hazard dynamics, similar to those introduced in Sect. 2.1.

Direct physical damage over time $t$ to an asset or infrastructure $j$ caused by concurrent impacts can therefore be simply calculated according to Eq. (10).

$$d_j(t)[t_{S_1}, t_{REC_2}] = \begin{cases} f_2 & \text{if } t_{S_1} \leq t < t_{\text{RES}_1} \\ f_2 \cdot R_{j,1}(t) & \text{if } t_{\text{RES}_1} \leq t < t_{\text{REC}_1} \\ 0 & \text{if } t_{\text{REC}_1} \leq t < t_{S_2} \\ f_4 & \text{if } t_{S_2} \leq t < t_{\text{RES}_2} \\ f_4 \cdot R_{j,2}(t) & \text{if } t_{\text{RES}_2} \leq t < t_{\text{REC}_2} \end{cases} \tag{10}$$

with

$$f_2 = f_{v_{j,1}}(h_{1,\max}) \tag{11}$$

$$f_4 = f_{v_{j,2}}(h_{2,\max}) \tag{12}$$

According to Eq. (10), at time $t_{S_1}$, the first hazard event causes the asset $j$ to sustain damage quantified as $f_2$. Once the response phase of the first event concludes (i.e., for $t \geq t_{\text{RES}_1}$), the asset begins to recover from the damage using the recovery function $R_{j,1}(t)$. By $t = t_{\text{REC}_1}$, the asset has fully recovered, resulting in a residual damage level of zero, which remains unchanged until a second hazard occurs at $t_{S_2}$.



At $t_{S_2}$, the second hazard affects the asset, causing damage quantified as $f_4$. Similarly, once the response phase of this second event is completeed (i.e., for $t \geq t_{\mathrm{RES}_2}$), the asset begins to recover according to the recovery function $R_{j,2}(t)$. By $t = t_{\mathrm{REC}_2}$, the asset again reaches a residual damage level of zero. Equations (11) and (12) describe single-hazard damage models as functions of the maximum intensities of hazard events, $h_{1,\mathrm{max}}$ and $h_{2,\mathrm{max}}$, respectively. Consistent with the single-hazard scenario (Sect. 2.1), it is assumed that the damages from both events occur entirely at the onset of each event. This evolution of damage in

time, as described in Eq. (10), is graphically represented in Fig. 3, panel (c).

Although it may seem that the impacts in this scenario could be assessed by treating the two events as independent single hazards, the situation is more complex. In an ideal analytical scenario, the system's vulnerability may have either decreased or increased after reconstruction following the first event. Some studies suggest that both structural and non-structural attributes can improve in response to previous events, as seen in the "built back better" theory (Kennedy et al., 2008; Neeraj et al., 2021).

Consequently, a second event with similar characteristics could cause more or less damage, depending on how the recovery and reconstruction process unfolded after the initial response phase. Given this scenario, it becomes crucial for risk assessment and management to update the system's vulnerability and exposure after the recovery process to accurately calculate the risks posed by consecutive hazards.

## 2.3    Generalised Damage Framework

The three distinct types of impact interactions, as defined by Eqs. (5), (7), and (10), are unified into a single formulation, as presented in Table A1. This generalised piecewise defined function allows the estimation of relative damage over time to an asset or infrastructure $j$ exposed to $n$ natural hazards. The parameters $\Theta$ and $\mu$, as defined in Eqs. (3) and (4), govern the classification of impact interactions:

–    $\Theta = 0$: Impacts are classified as *concurrent*, and formulation, presented in Table A1simplifies to Eq. (5).

–    $\Theta = 1, \mu = 0$: Impacts are classified as *consecutive*, and formulation, presented in Table A1 simplifies to Eq. (7).

–    $\Theta = 1, \mu = 1$: Impacts are classified as *independent*, and formulation, presented in Table A1 simplifies to Eq. (10).

The framework has been implemented in Python to facilitate its practical application. The Python code provides a modular structure, enabling users to easily customise multi-hazard parameters (e.g., intensity, duration, and temporal overlap of the events), vulnerability functions, and recovery dynamics. The modularity of the Python implementation enhances flexibility by

allowing straightforward integration with existing hazard analysis workflows, scalability across multiple scenarios and case studies, and adaptability to user-defined data formats and external modelling tools. For further details on the code, see Sect. *Code and data availability* at the end of the manuscript.

The core of the framework is the recovery function $R_{j,i}(t)$, which models the temporal dynamics of response and recovery for each hazard. Recovery rarely follows a uniform trajectory, as it is shaped by socioeconomic, infrastructural, and political

factors. Linear recovery, for instance, assumes a constant rate of progress over time, suitable for regions with predictable and stable resource availability. Exponential recovery captures scenarios where initial efforts are rapid but diminish as resources are exhausted, often observed in regions with strong emergency response but limited sustained capacity.



More complex recovery dynamics can be modelled using logistic functions, particularly useful to describe real-world conditions in which initial delays are followed by a period of rapid progress once external support or efficiency gains come into play.
Generically, the framework supports any custom recovery functions, enabling users to tailor their models precisely to specific contexts.

## 3 Puerto Rico Case Study and Multi-Hazard Damage Assessment

This section aims to demonstrate the applicability and added value of the proposed generalised damage framework in capturing the complex, cumulative effects of temporally compounding hazards. Puerto Rico offers a highly relevant and well-documented
case, having experienced a sequence of major disasters within a relatively short time window. The consecutive impacts of Hurricanes Irma and Maria in 2017, followed by the 2019-2020 earthquake sequence, underscore the need for multi-hazard damage assessment approaches that go beyond traditional single-hazard models. This case study offers a real-world example to underline how multi-hazard interactions, recovery trajectories, and vulnerability evolution can meaningfully influence the outcomes of a disaster sequence.

In contrast to single-hazard risk assessments, multi-hazard damage modelling poses specific challenges at each stage of the analysis, from hazard interaction modelling and exposure characterization to the representation of dynamic vulnerability and the quantification of cumulative impacts. The spatial and temporal interdependencies among hazards may significantly alter the nature and magnitude of damage. The Puerto Rico case study provides a compelling example of how neglecting these interactions can lead to an underestimation of systemic risk, overlooking how prior damage and interrupted recovery processes
influence vulnerability to subsequent events.

In September 2017, Puerto Rico was struck by two Category 5 hurricanes, Irma and Maria, within just a few weeks. These events caused catastrophic damage to housing, lifelines and public infrastructure (Amir et al., 2020; Morales-Velez et al., 2021; Kwasinski et al., 2019; Cangialosi et al., 2018; Pasch et al., 2023; Fischbach et al., 2020). Hurricane Maria, in particular, caused nearly 3,000 deaths and an estimated \$90 billion in damage, ranking as the third most expensive hurricane in US
history (Kishore et al., 2018; FEMA, 2018a). Recovery efforts were slow and uneven; of the \$42 billion allocated by Congress, only \$14 billion had been disbursed by the end of 2019 (FEMA, 2019a; GAO, 2020). As a result, large portions of critical infrastructure remained vulnerable when a magnitude 6.4 earthquake struck on 7 January 2020. This earthquake, part of a wider seismic sequence that began in late 2019, exacerbated the vulnerability of an already weakened territory (Melendez-Colon et al., 2021; Hu et al., 2019). More than 300 schools were reported to have suffered structural damage, and more than
8,000 residents were displaced (Rodriguez et al., 2021). The disaster further destabilised the power grid and impeded access to essential services, especially in southern municipalities. The healthcare sector, already strained, faced growing operational challenges (Peters et al., 2021), while the tourism and agricultural sectors registered significant economic losses (Puerto Rico Chamber of Commerce, 2020).

Figure 4 illustrates the sequence and overlap of the two major types of hazards that affected Puerto Rico during the 2017-
2020 period.




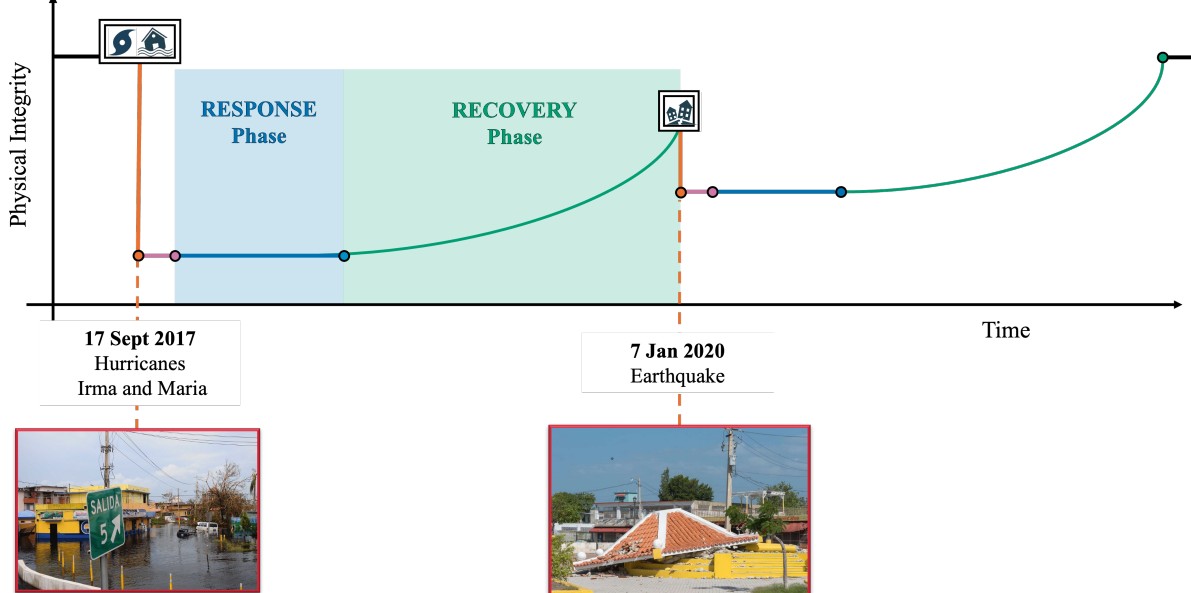

**Figure 4.** Representation of Puerto Rico's 2017-2020 disaster sequence. Images, from left to right: (i) Flooded area in Carolina, Puerto Rico, following Hurricane Maria's impact on the island, 29 September 2017. Photo by Sgt. Jose Ahiram Diaz-Ramos. Online image, Flickr. (ii) Severe earthquake damage to a gazebo in a public park in Guánica, Puerto Rico, 11 February 2020. Photo by Liz Roll/FEMA. Online image, NARA & DVIDS Public Domain Archive.

Puerto Rico is a densely inhabited island, characterised by a higher concentration of buildings in the San Juan metropolitan area, adjacent northern municipalities, and in the southern region around Ponce. These areas not only host a large share of the residential, commercial, and public infrastructure, but also represent key nodes in the island's economic and logistical networks. The multi-hazard damage assessment performed applying our framework focuses on damages to the built-up area, using as input a portfolio of buildings provided by FEMA (2020). This portfolio was compiled as part of a recent data inventory conducted on the island after Hurricane Maria, and includes the following categories, along with their corresponding estimated replacement values FEMA (2020):

- **Residential**: $\sim 1.5 \times 10^6$ buildings; estimated replacement value $\sim 300$ billion USD.

- **Commercial**: $\sim 6 \times 10^3$ buildings; estimated value $\sim 12$ billion USD.

- **Industrial**: $\sim 1 \times 10^3$ buildings; estimated value $\sim 2$ billion USD.

- **Educational**: $\sim 1.2 \times 10^3$ buildings; estimated value $\sim 3$ billion USD.

The analysis is carried out in two phases. In the first, we model the impacts of Hurricane Maria using a Compound Damage Model (Sect. 2.2.1), integrating wind and flood hazards. The fragility curves used for this phase are those provided by FEMA for the specific context of Puerto Rico (Federal Emergency Management Agency, 2020). In the second phase, the Consecutive





Damage Model (Sect. 2.2.2) is applied to simulate the seismic impacts on structures already weakened by hurricane damage. Here, we started from the standard fragility curves provided by FEMA, and we modified them to capture the changed vulnerability of assets previously affected by the hurricane, and still exhibit residual damage.

Input data includes:

- **Structural parameters**: Material, type, number of stories, code compliance.

- **Hazard inputs**: Wind speed, flood depth, spectral acceleration.

- **Vulnerability functions**: Hazus-based curves, modified to reflect state dependency.

- **Economic indicators**: Repair cost per unit area, downtime, and replacement costs.

The results are expressed in terms of physical loss (reduction in structural integrity) and economic cost.

### 3.1 Compound Hurricane Impacts: Wind and Flood Interaction

Within the context of Puerto Rico's 2017 disaster sequence, the compound impacts of Hurricane Maria represent a particularly critical component of the multi-hazard damage landscape. As highlighted in the preceding sections, the spatial distribution of exposure and the intensity of the event created conditions for widespread disruption. This subsection focuses on the estimation of the damages caused by the interaction between two cooccurring hazards: extreme winds and flooding.

According to our framework, these damages are assessed using a Concurrent Damage Model (Sect. 2.2.1). The vulnerability
functions applied are those provided by the FEMA Hazus-MH methodology FEMA (2020). Hazus-based fragility curves for wind damage are defined according to structural characteristics such as primary construction material, roof anchorage, and building typology. Similarly, flood damage is modelled using regionally calibrated depth-damage functions, which express the probability of structural damage based on the depth of the inundation. In the case of Hurricane Maria, the synergistic nature of these hazards became particularly evident: high wind speeds contributed to roof detachment and building envelope
failures, while subsequent or simultaneous flooding infiltrated already compromised structures. FEMA after-action assessments reported that buildings affected by overlapping hazards showed repair costs up to 30–40% higher than those affected by wind or flood alone FEMA (2018b). Field observations highlighted that flood depths exceeding 1.5 metres resulted in near complete failure of unreinforced masonry buildings, particularly where prior wind damage had already compromised structural elements Rodriguez et al. (2021).

The simulation results indicate that the combined impacts of wind and flood during Hurricane Maria resulted in approximately $90 billion in damage in Puerto Rico. This estimate is consistent with official reports, which attribute nearly $90 billion in total damage to the effects of the hurricane on the island National Oceanic and Atmospheric Administration (2018).

### 3.2 The Recovery Process and Subsequent Seismic Impacts

The earthquake sequence that began in late 2019 and culminated in the magnitude 6.4 event on 7 January 2020 struck Puerto
Rico at a time when many regions were still in the process of recovering from Hurricane Maria. This temporal overlap be-



tween the recovery phase and the occurrence of a new hazard event represents a critical scenario within multi-hazard damage assessment, where pre-existing damage and delayed reconstruction substantially influence the vulnerability landscape.

To spatially contextualise this interaction, Fig. 5 integrates three key spatial data sets: the track of Hurricane Maria (September 2017), the distribution of buildings still damaged as of November 2018 (based on Hazus post-event assessments), and the

contours of the peak ground acceleration (PGA) from the January 2020 earthquake. The post-hurricane assessment identified nearly 138,000 buildings in Puerto Rico that had not yet been fully repaired more than a year after the hurricane.

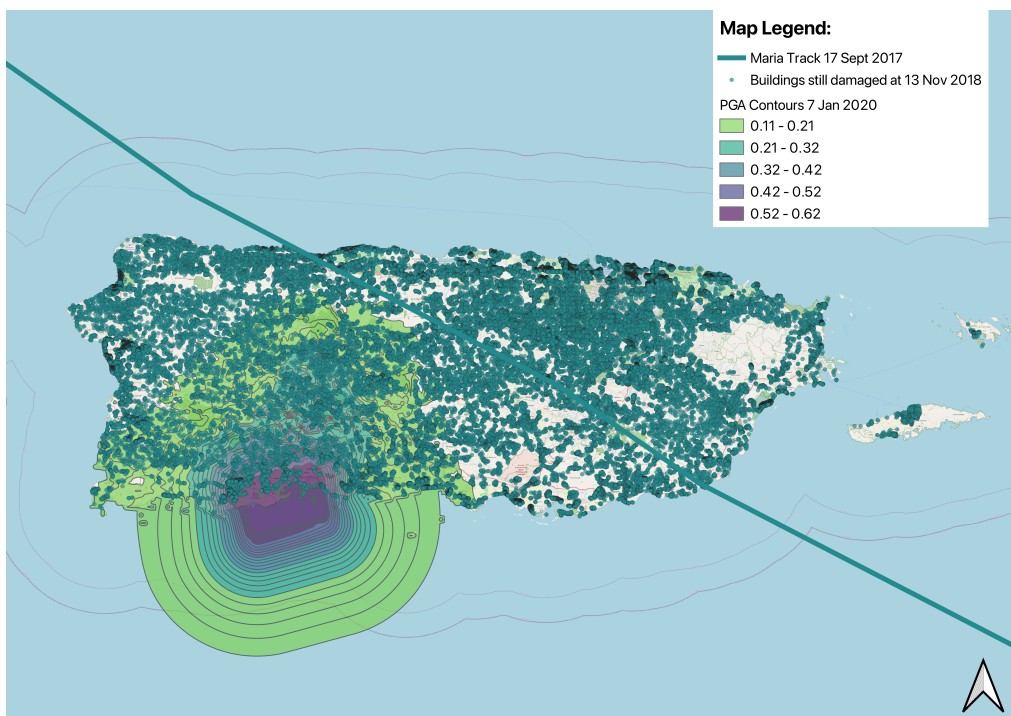

**Figure 5.** Map showing the track of Hurricane Maria (Sept. 2017), the spatial distribution of buildings with remaining hurricane damage (as of Nov. 2018), and PGA [g] contours from the January 2020 earthquake.

As illustrated in Fig. 5, many of the damaged buildings were concentrated in coastal and southern municipalities, including regions such as Ponce and Guayanilla, which later experienced the highest levels of ground shaking during the earthquake (PGA > 0.5g). These areas faced dual exposure, first to high winds and flooding from Hurricane Maria and later to seismic

loading from the 2020 event. Residual structural weaknesses, whether due to incomplete reconstruction or substandard repairs, likely increased the risk of damage and collapse.

To model the seismic impacts on the buildings still damaged by the hurricane, we applied a Consecutive Damage Model (Sect. 2.2.2). We started from the standard fragility curves provided by FEMA (Federal Emergency Management Agency, 2012), and we transformed them into state-dependent curves, to properly account for residual damage. Our approach introduces





parametric adjustments to account for pre-existing damage conditions. Specifically, the median values of both structural and non-structural fragility curves are shifted according to hypothetical levels of residual damage.

Given the absence of detailed spatialised building-level recovery data, we assumed a homogeneous distribution of residual damage throughout the building portfolio. Furthermore, since no specific residual damage values were available, we simulated different degrees of recovery to retrospectively identify those that best captured the final estimated losses.

To explore these variations, we implemented a sensitivity analysis in which structural and non-structural fragility curves are adjusted by increments of 5% to 30%. This approach allowed us to estimate how the results of the damage could vary under different assumptions about the state of the building stock at the time of the earthquake. Although the modifications were applied uniformly, their impact was particularly pronounced in areas with high PGA values, where even small increases in vulnerability result in disproportionately large increases in expected losses.

Figure 6 illustrates the conceptual effect of residual damage on fragility curves. As residual damage increases, the curves change to the left, reflecting a greater susceptibility to damage at lower levels of seismic demand.

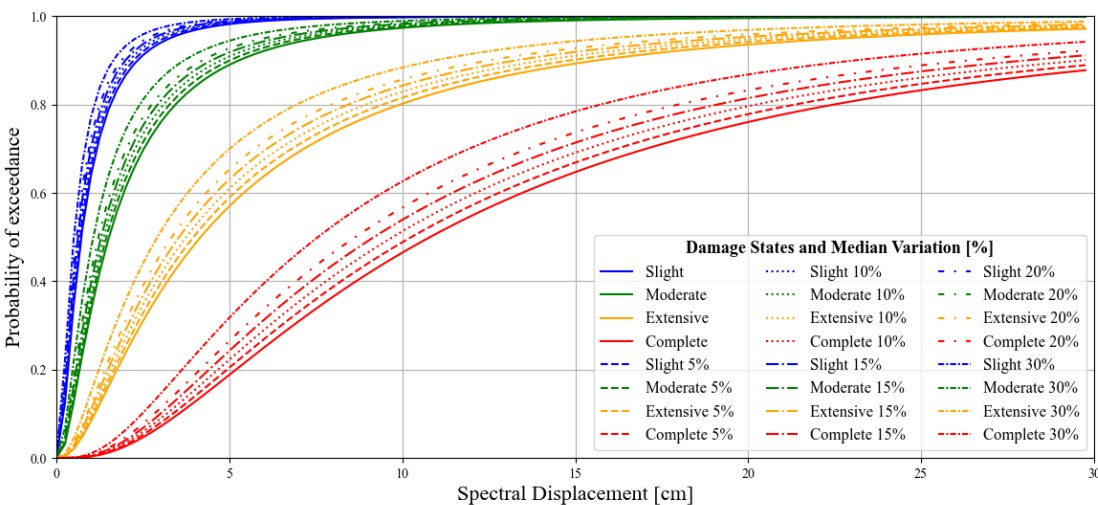

**Figure 6.** State-dependent fragility curves reflecting changes in structural vulnerability due to residual damage.

The results of this analysis are presented in Table 2. The matrix shows total estimated damages (in thousands of USD) for combinations of structural and non-structural degradation. We obtained these values by translating the simulated physical damage into economic losses, following the damage-to-loss conversion methodology described in the Hazus-MH Technical

Manual (FEMA, 2020). For each type of building and damage state, predefined loss ratios were assigned, representing the expected repair cost as a percentage of the replacement value of the asset. These ratios differ for structural and non-structural components and increase with the severity of damage. The replacement cost values used in this analysis correspond to the





exposure $E_j(t)$ provided in the FEMA asset inventory. This approach ensures a consistent and standardised translation from simulated damage to direct economic loss across the entire building portfolio.

A clear non-linear trend emerges from the resulting loss estimates: as both forms of residual damage increase, the total losses escalate sharply. For instance, at 10% structural degradation and 15% non-structural degradation, losses rise by over 40% compared to the baseline (undamaged) scenario.

**Table 2.** Total damage estimates (in thousands of $) for different combinations of fragility curve modifications.

|  | NonStruct % 0 | NonStruct % 5 | NonStruct % 10 | NonStruct % 15 | NonStruct % 20 | NonStruct % 30 |
|---|---|---|---|---|---|---|
| **Struct % 0** | 1'205'525 | 1'316'141 | 1'446'902 | 1'602'471 | 1'789'239 | 2'294'623 |
| **Struct % 5** | 1'245'367 | 1'355'773 | 1'486'192 | 1'641'565 | 1'828'054 | 2'331'851 |
| **Struct % 10** | 1'292'172 | 1'402'442 | 1532614 | 1'687'487 | 1'873'685 | 2'376'321 |
| **Struct % 15** | 1'347'889 | 1'457'732 | 1'587'716 | 1'742'296 | 1'927'840 | 2'429'457 |
| **Struct % 20** | 1'414843 | 1'524'206 | 1653'575 | 1'807'901 | 1'933'072 | 2'493'172 |
| **Struct % 30** | 1'595'392 | 1'703'806 | 1'832'623 | 1'985'067 | 2'168'596 | 2'934'148 |

To better visualise this increase in losses, Fig. 7 presents a three-dimensional surface plot. The curvature of the surface underscores the accelerating cost of residual damage, especially in scenarios where structural and non-structural elements 420    remain damaged.





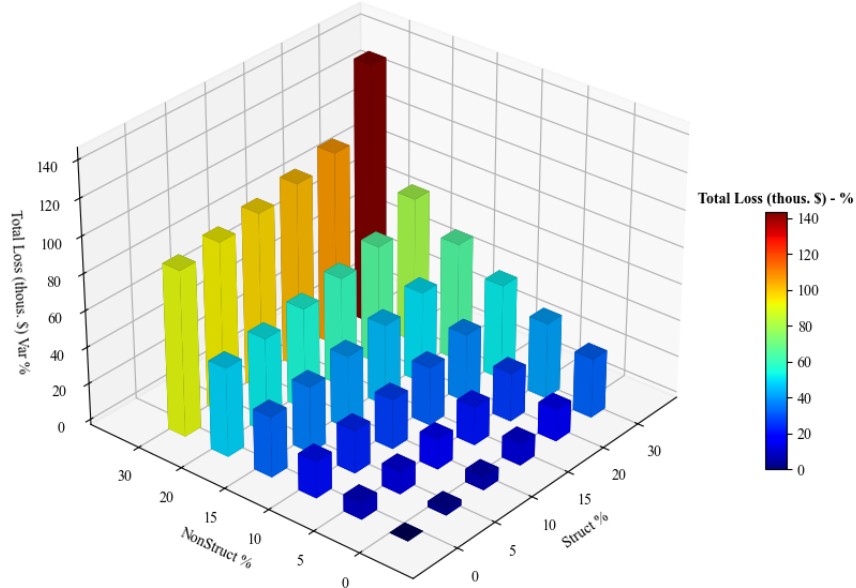

**Figure 7.** Three-dimensional representation of total losses (in thousands of $) for different combinations of structural and non-structural fragility curve modifications.

The official damage estimates provided by the Government of Puerto Rico (2020) amount to 1,700 thousand $. By introducing state-dependent fragility curves with a 15% adjustment for structural and non-structural damage components, we obtained a total simulated loss of 1,742 thousand $, closely matching the official estimate. In contrast, applying standard fragility curves, without accounting for residual damage, yields a simulated loss approximately 30% lower than the official figure. A comparison of the official estimates and those obtained with these two modelling configurations is presented in Table 3.

From a technical standpoint, this methodology offers a pragmatic solution for capturing the effects of incomplete recovery in multi-hazard settings. By adjusting the standard fragility functions originally developed in the Hazus framework to reflect state-dependent vulnerability, the model bridges the gap between sequential hazards and evolving exposure conditions. Although the analysis is limited by spatial resolution and the assumption of uniform adjustment, it provides valuable information on how delayed recovery can dramatically amplify future losses.



**Table 3.** Total damage in thousands of $ caused by the seismic sequence of 2019-202 in Puerto Rico: comparison between official estimates and modelling outcomes.

| Source | Total damage [Thous. $] |
|---|---|
| Official Portal of the Government of Puerto Rico (Government of Puerto Rico, 2020) | 1.700 |
| Modelled damage using state-dependent fragility curves *(consecutive hazards damage model)* | 1.742 |
| Modelled damage using standard fragility curves *(single hazard damage model)* | 1.206 |

For state-dependent fragility curves application, we refer to the implementation of a fragility curve modification of 15% for structural and non-structural damage components (see Sect. 3.2).

## 4 Discussion

This research introduces a generalised framework for quantitatively assessing multi-hazard direct physical damage to exposed assets, such as buildings or critical infrastructures, over time. It provides scientists and decision-makers with a comprehensive and deeper understanding of the impacts caused by compound and consecutive events. The framework is operationalised as

an open-source Python code (please refer to Sect. *Code and data availability* at the end of the manuscript), allowing the easy application of various users in a wide range of case studies and applications. It comprehensively covers damage mechanisms resulting from both concurrent and consecutive hazard events by modelling the increased damage resulting from the combined impact of two or more concurrent hazards, and the effect of cumulative damage and recovery dynamics in case of consecutive hazards. By offering a hazard-independent formulation, the framework can be applied to any combination of sudden-onset

hazard typologies. Furthermore, it allows for the seamless integration of available multi-hazard damage models, such as vector-valued or state-dependent fragility models (Gentile et al., 2022).

The efficacy of the framework is demonstrated through a real-world case study in Puerto Rico (Sect. 3), where both damage mechanisms have been observed: the concurrent impacts of wind and flood generated by the passage of Hurricane Maria, as well as the consecutive impacts caused by the subsequent seismic sequence of 2019-2020. Following the computation of

hurricane-induced losses, the framework is used to model the damages incurred by the consecutive effects of the hurricane and earthquake on a portfolio of buildings. To achieve this, a consecutive damage model is applied, involving the development of seismic state-dependent fragility curves. This model adjusts fragility function parameters for both structural and non-structural components, accounting for the conditional damage state of assets due to incomplete recovery from hurricane-induced damage. Multi-hazard damage is evaluated in both physical (e.g., the percentage loss of physical integrity) and economic terms (e.g.,

monetary loss). The application of a consecutive damage model better estimates real losses. By introducing state-dependent fragility curves, the total damage for the entire portfolio of buildings better captures the order of magnitude of the official total economic damage estimates for the earthquake, as depicted in Table 3.

These results provide user evidence of the need for more complex damage models that can account for interaction mechanisms between hazards, vulnerabilities, and impacts. From a policy perspective, they underscore the importance of prioritis-



ing rapid and equitable recovery, particularly in high-exposure zones, while also highlighting the necessity of multi-hazard-
       informed planning frameworks that account for the temporal dynamics of vulnerability.

       To enhance such assessments, future efforts should focus on coupling this modelling approach with finer-resolution recovery
       data, to enable more targeted and locally nuanced assessments. However, this study represents a concrete step forward in
       operationalising the concepts of cascading risk and temporally dynamic vulnerability in the context of compound hazard
events.

### 4.1    Assumptions and limitations

The development and application of this framework in the context of the Puerto Rico case study are subject to various as-
sumptions and limitations that inevitably introduce uncertainty but also define the scope and applicability of the study. These
assumptions and limitations, previously mentioned in various sections of this manuscript, are further detailed below to provide
clarity and transparency regarding the boundaries of this analysis.

***Focus on sudden-onset hazards***. The framework primarily targets sudden-onset hazards, characterised by short-duration
events such as hurricanes and earthquakes, where the event phase is brief compared to the subsequent response and recovery
phases. This assumption is particularly relevant for the Puerto Rico case study, where Hurricane Maria and the 2019-2020
earthquake sequence are treated as discrete events with well-defined temporal boundaries. However, this focus excludes long-
onset hazards such as droughts, sea-level rise, or subsidence, which evolve over extended periods and may blur the distinctions
between the event, response, and recovery phases (Terzi et al., 2022). Although this limitation does not diminish the appli-
cability of the framework to sudden-onset events, it underscores the need for separate modelling approaches for prolonged
hazards.

***Simplification of hazard interactions***. The framework models hazard interactions through state-dependent fragility curves
that adjust building vulnerability based on residual damage from previous events. While this approach provides a robust means
of capturing cumulative damage, it simplifies certain complexities inherent in real-world interactions. For example, the over-
lapped impacts of flooding and wind during Hurricane Maria were modelled as independent contributions to total damage.
However, field observations suggest that wind-induced roof damage often exacerbated flooding by allowing water infiltration
into buildings, creating a compounding effect (FEMA, 2020). These nuances are not fully represented in the current imple-
mentation.

***Direct physical impacts only***. The framework is designed to estimate direct physical damages to assets, such as the percent-
age of structural integrity lost, using vulnerability or fragility functions as represented by Eqs. (6), (9), (11), and (12). While
these functions are effective for quantifying physical impacts, they do not capture indirect effects such as business interruptions,
supply chain disruptions, or social impacts. In Puerto Rico, the prolonged loss of electricity and water after Hurricane Maria
led to widespread socioeconomic challenges, including health crises, migration, and economic decline in key sectors such as
agriculture and tourism (Fischbach et al., 2020; Peters et al., 2021). These indirect effects, while significant, fall outside the
scope of the current framework.



*Assumptions about the recovery process*. The recovery process is modelled as a series of linear or exponential functions representing slow, medium, or fast recovery speeds. This approximation facilitates analysis but oversimplifies the inherently
non-linear nature of recovery, which is influenced by a multitude of factors. In Puerto Rico, for example, bureaucratic inefficiencies, delayed federal funding, and inequitable distribution of resources significantly delayed recovery efforts in certain regions (Group, 2020; OCHA, 2020). Moreover, the framework assumes uniform recovery rates across structural and non-structural components, which may not reflect reality. Non-structural elements, such as utilities and interior finishes, often recover faster than structural systems, leading to mismatches in the actual recovery pace (FEMA, 2019b).

*Simplification of temporal dynamics*. In the framework, it is assumed that physical integrity is lost immediately at the onset of the event, and no further degradation occurs during the response phase. While this assumption is practical for mathematical modelling, it does not capture cases where damage continues to accumulate after the hazard peaks, such as when flooding persists or secondary hazards occur. In addition, physical integrity is assumed to remain constant during the response phase, despite minor fluctuations that may occur due to emergency repairs or further deterioration. Although these simplifications do
not introduce significant bias, they limit the ability to model more complex temporal dynamics observed in real-world scenarios (Porter et al., 2020).

*Generalized spatial resolution*. The framework adopts a portfolio-based approach, grouping buildings into categories based on general characteristics such as structural type, construction material, and occupancy use. While this approach is suitable for regional-scale assessments, it does not capture localised variations in vulnerability or hazard exposure. In Puerto Rico,
differences in topography, proximity to fault lines, and floodplain dynamics played a significant role in shaping the spatial distribution of damage (Baker et al., 2021; FEMA, 2020). These localised effects were approximated rather than explicitly modelled, potentially reducing the precision of damage estimates.

*Economic impact modelling limitations*. The economic losses in the Puerto Rico case study were estimated using standard unit repair costs, downtime estimates, and sector-specific multipliers. Although these methods provide a useful approximation,
they do not account for broader macroeconomic effects, such as long-term population decline, loss of workforce productivity, or disruptions in international trade. For example, the loss of agricultural infrastructure during Hurricane Maria had cascading effects on food security and export revenues, which are not explicitly captured in the framework (Fischbach et al., 2020; Group, 2020).

*Uncertainty in state-dependent fragility curves*. Adjustment of fragility curves to reflect residual damage is based on propor-
tional modifications to the median curve values. Although this method effectively accounts for cumulative damage, it assumes uniform vulnerability changes across all buildings within a category. In reality, building-specific factors such as construction quality, maintenance history, and localised damage patterns may lead to significant variability in vulnerability (Nofal and van de Lindt, 2020; Porter et al., 2020). This limitation highlights the need for more granular data to refine the state-dependent fragility model.

In conclusion, while the framework provides a robust and adaptable tool for modelling multi-hazard impacts, its assumptions and limitations must be carefully considered when interpreting the results of the Puerto Rico case study. Addressing these limitations in future iterations, such as incorporating dynamic recovery models, indirect impacts, and higher spatial resolution,



will improve the accuracy and applicability of the framework, allowing more comprehensive assessments of multi-hazard scenarios.

## 4.2 Future developments

The proposed framework, while comprehensively addressing the assessment of direct physical damages caused by multiple hazards, presents opportunities for significant improvements. These advancements would not only improve the accuracy and applicability of the framework but also broaden its potential for integration into various fields of disaster risk reduction and management. In the following, we highlight several key directions for future developments.

***Better understanding and modelling recovery dynamics***. The dynamics of the recovery process are central to the assessment of residual damage and the interactions between consecutive hazard impacts. Specifically, the timing of the response phase ($t_{RES}$) determines whether the hazard impacts are classified as concurrent or consecutive, while the timing and shape of the recovery phase ($t_{REC}$ and $R_j(t)$) significantly influence the residual damage for subsequent events. The response phase, while critical for civil protection and disaster preparedness, has received limited attention in the scientific literature, often being conflated with the recovery phase. However, its proper evaluation is essential to improve disaster safety plans and understand its cascading effects on recovery (UNISDR, 2015; Consiglio dei Ministri, 2018; IFRC, 2020; Toyoda et al., 2021). More attention has been paid to recovery dynamics, particularly in the context of seismic resilience. Recovery dynamics are highly variable, often following linear, exponential, or logistic patterns depending on socio-economic, political, and governance factors. The pioneering work of Miles and Chang introduced one of the first models of recovery behaviour after an earthquake, proposing recovery trajectories of the community (Miles and Chang, 2006). Based on this, Cimellaro et al. (2010) developed an advanced method for the quantification of resilience by integrating recovery and preparedness metrics. Recent studies have highlighted the importance of recovery models in post-disaster decision-making (Marasco et al., 2022; Fountain and Cradock-Henry, 2020; Loos et al., 2023), underscoring the need for reliable recovery functions. However, challenges remain in quantifying recovery due to the complexity of the influencing factors, including governance, technology, economic conditions, and social cohesion. Emerging methodologies, such as Agent-Based Models (ABMs), offer promising avenues to simulate recovery processes by integrating interactions between individuals, institutions, and resources (Xu and Chopra, 2022). These models complement traditional approaches like community surveys and questionnaires, which are invaluable for understanding local recovery dynamics and identifying barriers to resilience (Jones and Ballon, 2020; Opabola et al., 2023).

***Providing more reliable multi-hazard damage models***. Improving multi-hazard damage models is essential for capturing the cumulative effects of consecutive hazards. Future research should focus on integrating damage-state-dependent functions that dynamically modify vulnerability and fragility curves based on residual damage from prior events. For example, scenarios involving hurricanes followed by earthquakes, as demonstrated in the Puerto Rico case study, require models capable of adapting to changes in structural and non-structural vulnerabilities caused by sequential impacts (Porter et al., 2020; Nofal and van de Lindt, 2020).

***Evaluating dynamic exposure over time***. Exposure is a dynamic parameter that evolves as a result of maintenance, retrofitting, and functional changes in assets. Hazard impacts can reduce exposure through physical damage, but exposure may increase




following recovery processes or new construction. Future developments should aim to model these variations over time, capturing the cyclical nature of exposure changes in response to hazards and human interventions (Gomez-Cunya et al., 2022; Loos et al., 2023).

***Incorporating indirect impacts through functionality modelling***. The framework currently focuses on direct physical damage but does not explicitly address indirect impacts, such as loss of functionality, economic disruption, or cascading failures in interconnected systems. Future iterations could incorporate functionality as a key metric, using physical integrity as a proxy to estimate functionality (Miles et al., 2019; Opabola et al., 2023). This approach would enable assessments of broader systemic impacts, including business interruptions and supply chain disruptions.

***Adapting the framework for long-onset hazards***. The current framework is optimised for sudden-onset hazards, where event durations are short relative to the response and recovery phases. However, long-onset hazards such as droughts, coastal erosion, and subsidence require a fundamentally different modelling approach. Extending the framework to account for these dynamics would improve its applicability to a wider range of hazard scenarios (Terzi et al., 2022).

    By addressing these areas, the framework can evolve into a more comprehensive tool for multi-hazard risk assessment and
management. These advances would support decision-makers in developing more effective mitigation strategies, improving community resilience, and optimising resource allocation during the recovery process.

## 5   Conclusions

This paper presents a generalised mathematical framework for assessing direct physical damage to exposed assets subjected to multiple natural hazards that can overlap or occur sequentially over time. Unlike existing models that focus on specific asset-
hazard combinations, our framework is designed to be broadly applicable to multiple hazards, capturing the complexities of concurrent, consecutive, and independent hazard interactions. A key contribution of this work is the incorporation of recovery dynamics into the damage assessment process. By formalising the changes in vulnerability between consecutive hazards, our approach allows for capturing post-disaster loss dynamics. The modular Python implementation of the framework ensures flexibility, enabling users to customise hazard parameters, vulnerability functions, and recovery processes and apply the framework
to their specific case study.

    Its effectiveness is demonstrated in a Puerto Rico case study, where it was applied to model damage from Hurricane Maria and the subsequent 2019-2020 seismic sequence. By incorporating state-dependent fragility curves to adjust asset vulnerability based on residual damage, the framework produced damage estimates that closely align with official economic loss assessments. This highlights the importance of accounting for hazard interactions and recovery dynamics to better capture
multi-hazard losses.

    Despite its contributions, the framework has limitations, particularly in introducing some simplifications in the hazard interactions modelling, assuming uniform recovery rates, and focusing solely on direct physical impacts while neglecting broader indirect socio-economic consequences. Future developments will address these challenges by refining recovery modelling, in-



corporating dynamic exposure changes, and expanding the applicability of the framework to long-onset hazards. In addition,
integrating functionally-based assessments and indirect economic losses will enhance its comprehensiveness.

*Code and data availability.* The Python implementation of the framework, along with detailed documentation, is freely available on GitHub
at the following link: https://github.com/AlessandroBorre/multi-hazard-impact-framework.git

.

## Appendix A:  Generalised Damage Formula

The generalised damage formula is summarised in Table A1, which shows how relative damage $d_j(t)$ evolves over time
depending on the type of hazard interaction (concurrent, consecutive, or independent), governed by the parameters $\Theta$ and $\mu$.

**Table A1.** Generalised damage formulation across time intervals for asset $j$

| Start Time | End Time | Damage Function $d_j(t)$ |
|:---:|:---:|:---:|
| $t_{S_i}$ | $\Theta \cdot t_{RES_i} + (1-\Theta) \cdot t_{S_{i+1}}$ | $\Theta \cdot f_2 + (1-\Theta) \cdot f_1$ |
| $t_{RES_i}$ | $t_{S_{i+1}}$ (only if $\Theta = 1$) | $\Theta \cdot f_2 \cdot R_{j,i}(t)$ |
| $t_{S_{i+1}}$ | $\Theta \cdot t_{RES_{i+1}} + (1-\Theta) \cdot t_{RES_{i,i+1}}$ | $\Theta \cdot \big(\mu \cdot f_4 + (1-\mu) \cdot \big(f_3 + [f_2 \cdot R_{j,i}(t_{S_{i+1}})] \cdot (1-f_3)\big)\big) + (1-\Theta) \cdot f_1$ |
| $\Theta \cdot t_{RES_{i+1}} + (1-\Theta) \cdot t_{RES_{i,i+1}}$ | $\Theta \cdot t_{REC_{i+1}} + (1-\Theta) \cdot t_{REC_{i,i+1}}$ | $\Theta \cdot \big(\mu \cdot f_4 + (1-\mu) \cdot \big(f_3 + [f_2 \cdot R_{j,i}(t_{S_{i+1}})] \cdot (1-f_3)\big)\big) \cdot R_{j,i+1}(t) + (1-\Theta) \cdot f_1 \cdot R_{j,i,i+1}(t)$ |

The generalised damage formulation presented in Table A1 provides a flexible framework for modelling how asset damage
evolves over time in response to multiple hazard events. By adjusting the parameters $\Theta$ and $\mu$, the model captures different
types of interactions, concurrent, consecutive, and independent, allowing for a consistent representation of complex multi-
hazard sequences.

*Author contributions.* A.B.: Conceptualization, Formal Analysis, Methodology, Investigation, Visualization, Writing – original draft prepa-
ration, D.O.: Conceptualization, Formal analysis, Methodology, Writing – review & editing, E.T.: Conceptualization, Writing – review &



editing, T.G.: Conceptualization, Writing – review & editing, G.Z.: Methodology, G.B.: Conceptualization, Writing – review & editing, S.D.A.: Conceptualization, Formal Analysis, Methodology, Investigation, Visualization, Writing – original draft preparation.

*Competing interests.* The authors declare that they have no conflict of interest.

*Acknowledgements.* We thank Dr. Roberto Rudari and Dr. Lorenzo Campo for their support in drafting a preliminary mathematical formulation for multi-hazard risks, which served as the foundation for the development of this work. We also thank Prof. Bruce D. Malamud for his insightful suggestions during an inspiring conversation at EGU24.





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
