# Peer review of "A mathematical framework for quantifying physical damage over time from concurrent and consecutive hazards"

_EGUsphere, 2025_

## Author Comment (AC1)

**REVIEWER 1**

**Major Concerns**

**Core Contribution and Scientific Value**

My primary concern relates to the fundamental claims about the manuscript's contribution and its practical utility. The authors position their work as providing a "generalised mathematical framework" for multi-hazard risk assessment, but this claim requires careful examination.

**Mathematical Formalization**: The mathematical framework presented is essentially a straightforward extension of existing single-hazard approaches, where total damage becomes a function of multiple hazards and recovery states. While the authors present this as novel, the mathematical formulation represents a straightforward extension. The real challenges in multi-hazard assessment lie not in the mathematical abstraction but in the empirical quantification of recovery functions, hazard dependencies, and vulnerability transitions over time - as the authors rightly state in sections 4.1 and 4.2.

**ANSWER**

We acknowledge that the mathematical framework represents an extension of existing single-hazard formulations, and we agree that its novelty lies less in the mathematical abstraction itself and much more in its **capacity to integrate multiple elements usually treated separately** (multi-hazard interactions, recovery dynamics, and multi-hazard vulnerability), and its **practical application**, **through the development of a dedicated Python code.**

In the revised manuscript, we will therefore reduce the emphasis on the formal mathematical aspects and clarify that the theoretical structure primarily serves as the foundation for a computational tool implemented in Python. To better reflect this focus, we will replace the term "mathematical framework" with "Python-based modelling tool" (or a similar term, as appropriate).

By strengthening these aspects, the revised manuscript will more accurately convey the originality of our contribution. It will position our work not as a purely theoretical and mathematical development, but as a practically

oriented, Python-based tool that operationalizes the relationships among multi-hazard overlaps, vulnerability dynamics, and recovery processes.

**Operational Utility Limitations:** More critically, the framework suffers from a fundamental limitation that severely constrains its practical applicability. The Puerto Rico case study illustrates this problem clearly. The authors demonstrate how Hurricane Maria affected earthquake vulnerability by using post-hurricane damage data to calibrate vulnerability adjustments. However, this approach is inherently backward-looking and requires empirical damage data that would not be available for forward-looking risk assessments.

**ANSWER**

We agree with the reviewer that the Puerto Rico case study mainly demonstrates a retrospective (ex-post) application based on post-event damage data. Nevertheless, we think that the proposed model can also be applied for anticipatory/ planning-oriented analyses.

In the revised version of the manuscript, we will explicitly introduce two modes of application within the same Python-based tool:

- **Forensic use:** ex-post analysis of consecutive events, where empirical data on damage and vulnerability are available. The Puerto Rico case study falls under this category and serves as a real-world example where we can compare two sets of damage curves and economic loss estimates, validating how earthquake fragility evolves after hurricane-induced damage. This demonstrates that the tool can reproduce physically observed effects without relying on the invention of new fragility curves.
- Anticipatory/ planning-oriented use: we will streamline the description of the mathematical framework and focus more on demonstrating how the model "behaves" under varying input parameters, specifically illustrating how total damage evolves as recovery, exposure, and vulnerability parameters change. This forward-looking approach enables sensitivity analyses that are valuable for preventive planning and the assessment of mitigation strategies, thereby better supporting anticipatory applications of the model.

To improve clarity, we will add a schematic figure summarizing these two operational pathways (forensic and anticipatory) within the same framework, showing how users can transition from retrospective validation to forward-looking assessments.

**Consider a practical scenario**: if a hurricane were to strike Puerto Rico tomorrow, practitioners using this framework would need to wait for post-hurricane damage assessments before they could adjust earthquake vulnerability functions. This severely limits the framework's utility for operational risk management, emergency planning, or prospective risk assessment. The authors do not adequately address how practitioners would estimate vulnerability changes in real-time or predictive applications without extensive post-event calibration data.

**ANSWER**

We would like to clarify that the objective of the Puerto Rico case was not to calibrate recovery or vulnerability functions for "predictive" purposes, but to use a real-world dataset where observed damages were available to validate the internal behaviour of the model. This forensic validation demonstrates that the tool can reproduce physically consistent variations in fragility following sequential hazards, without introducing arbitrary new curves.

In the revised manuscript, we will more clearly present the Puerto Rico case as a validation of the model's forensic capabilities. We will also clarify how, beyond this validation, the same framework can support forward-looking (i.e., anticipatory/ planning-oriented) analyses by simulating future scenarios through parameterized adjustments of recovery, exposure, and vulnerability functions. This will help illustrate the tool's relevance both for retrospective studies of past events and for forward-looking applications in disaster risk management.

**Misalignment of Claimed vs. Actual Contributions:**

The manuscript lists several important developments as "limitations" (section 4.1) or "future research directions" (section 4.2) that would actually constitute the real scientific advances needed in multi-hazard assessment. These include empirically-derived vulnerability transition functions, standardized recovery curves based on extensive post-disaster data, and predictive models for hazard-induced vulnerability changes. Only after solving these challenges would a general mathematical framework provide meaningful operational value.

Specifically, I was excited to see the authors propose an approach to adjust vulnerability curves to account for pre-existing damage conditions (lines 394-396). Such an approach represents a potentially valuable contribution to the field. However, the implementation still relies entirely on post-event calibration data, making it neither operational for forward-looking assessments nor generalizable across different contexts without extensive empirical datasets.

In conclusion, current framework appears most suited for retrospective analysis and systematic post-disaster impact assessment rather than the forward-looking risk management applications that the authors suggest. This represents a significant gap between the claimed contribution and the demonstrated capabilities.

**ANSWER**

We fully agree that empirically-derived vulnerability transition functions, standardized recovery curves, and predictive models for hazard-induced vulnerability changes are critical for achieving fully operational, forward-looking applications. These represent important avenues for future research and development, and we have clearly marked them as limitations or future directions in Sections 4.1 and 4.2.

We would like to clarify that the primary objective of the current work is not to deliver a fully operational anticipatory/ planning-oriented tool, but rather to provide a flexible, Python-based framework capable of integrating multi-hazard, recovery, and dynamic vulnerability components in a consistent, modular way.

**In the revised manuscript, we will:**

- 1. Emphasize that the framework is intended primarily for retrospective validation and sensitivity analyses rather than real-time operational deployment.
- 2. Highlight the novelty and added value of the tool in integrating components often treated separately in the literature (multi-hazard, recovery, dynamic vulnerability) and providing a Python-based tool that can support both forensic validation and anticipatory scenario exploration.

This clarification will help align the presentation of the framework with its actual demonstrated capabilities, while still pointing to its potential for future improvements.

**Language and Presentation**

The manuscript suffers from clarity and communication issues that hinder comprehension of the technical content. The writing style relies on unnecessarily complex sentence structures that obscure rather than illuminate key concepts. Many sentences contain multiple subordinate clauses that could be simplified without losing technical precision.

**ANSWER**

We will revise the whole manuscript and simplify the sentence structure, avoiding as much as possible the use of subordinate clauses.

Additionally, the manuscript exhibits significant redundancy, with core concepts repeated across sections without advancing the argument or providing new information. For example, the distinction between "concurrent and consecutive hazards" is mentioned repeatedly in the abstract, introduction, and methodology sections without substantive development of how the framework addresses each case differently.

**ANSWER**

We will significantly shorten the part devoted to the presentation of the mathematical framework, to dedicate more space to presenting how the model "behaves". This will help in reducing redundancy and concept repetition.

The technical exposition would benefit from more precise language and clearer logical flow. Terms like "generalised framework" are used extensively without clear definition of what makes the approach "general" compared to existing methods.

**ANSWER**

We will revise the language and ensure coherence and precision of the applied terminology. Moreover, as anticipated, we will avoid using the term "mathematical framework" and rather refer to a "Python-based tool" or similar.

**Recommendations for Improvement**

The manuscript addresses an important problem in disaster risk science, but it requires substantial revision to align claims with demonstrated capabilities. The authors should consider reframing their contribution more modestly and honestly. Rather than claiming a key contribution in operational multi-hazard assessment, they could position their work as a systematic methodology for post-disaster impact assessment or as a research template for understanding multi-hazard interactions in well-documented cases.

**ANSWER**

We thank the reviewer for this suggestion and agree that the manuscript should better align its claims with demonstrated capabilities. In the revised version, we will place **less emphasis on the mathematical formalization** and instead **highlight the Python-based framework** and its two complementary modes of application: (i) forensic and (ii) anticipatory/ planning-oriented.

This revised framing clarifies that the framework's **operational value lies in structured forensic analyses and scenario-based explorations**, while not implying real-time predictive applicability, and highlights the practical benefits of its Python-based implementation.

The framework has genuine value as a foundation for systematic vulnerability state tracking and post-disaster learning, but the authors should acknowledge its current limitations for predictive applications. Future work should focus on developing the empirical foundations needed to make such a framework operationally useful, including physics-based vulnerability transition models and standardized recovery parameters that can be estimated without extensive post-event data.

**ANSWER**

We fully agree that the framework is not applicable for real-time predictions. In the revised manuscript, we will explicitly acknowledge these limitations and clarify that the framework is intended primarily for forensic analyses and scenario-based forward-looking assessments. This reframing will ensure an honest representation of the tool's current capabilities while highlighting avenues for future advancement, including the development of a stronger empirical foundation for operational use, such as physics-based vulnerability

transition models and standardized recovery parameters that can be applied without relying on extensive post-event datasets. These aspects are already discussed in the "Future Developments" section of the manuscript.

The language and presentation issues require comprehensive editing to improve clarity and eliminate redundancy.

**ANSWER**

As anticipated, we will review the whole manuscript, simplifying the sentences, improving the coherence and precision of the adopted terminology, and avoiding unnecessary repetitions.

**Minor comments**

L30: "hazard" in brackets?

L 48: "underlined" seems to be a word choice error. Consider "highlighted", "identified", "outlined", "emphasized".

Line 105: "occurs integrally at the beginning" is unclear - suggest "occurs entirely at the beginning" or "occurs instantaneously at the beginning" to clarify that all damage happens at once rather than gradually.

L 110: To my knowledge, figures should be referenced in sequential order (Fig. 1, then Fig. 2, then Fig. 3, etc.) as they appear in the text. Fig. 3 is introduce before Fig. 1 and 2.

L 118, 128: Avoid describing figure elements by color ("blue line", "green line"). Use descriptive labels instead for accessibility and clarity (e.g., "the horizontal line representing the response phase"). Related and for figures in general: Ensure all figure elements are clearly labeled in the legend. Figure 3 legend: "The second event in temporal order" is redundant - "second event" already implies temporal sequence. Suggest removing "in temporal order" throughout.

L 179: Remove "fortunate" - scientific writing should avoid value judgments. L 279: Typo in "build" back better

L 407, Table 2: The values in Table 2 seem to be in USD, not "thousand USD". Please double check. Also check the formatting of the values in Table 2. Table 2/Figure 7: Both show the same results. Consider moving one of the two elements to the supplementary material. L 408: Line X: The damage-to-loss conversion methodology needs brief explanation rather than just a citation. Readers should understand the key assumptions without consulting external sources.

L 409: "Predefined loss ratios" needs clarification - predefined by whom, based on what data, and are they appropriate for Puerto Rico conditions? Specify the source and empirical basis.

L 415: The claim of "clear non-linear trend" needs quantitative support. How was non-linearity assessed? Provide statistical analysis (R², trend coefficients) rather than visual inspection and isolated examples.

L 421/423: "1,700 thousand" "1,742 thousand" is confusing and not scientific. Replace with scientific notation or standard units.

Throughout: Use standard terminology "slow-onset hazards" rather than "long-onset hazards" to align with established disaster risk literature (e.g., UNDRR, IPCC terminology).

**ANSWER**

We will carefully revise all these minor comments and integrate them into the revised version of the manuscript.

---

## Author Comment (AC2)

**REVIEWER 2**

**COMMENT 1**

First of all, I think that the term 'mathematical framework' feels a bit of a stretch. They are somewhat simple equations that are merely a mathematical formulation of the recovery curves as presented by De Ruiter et al. (2019) in the paper "Why we can no longer ignore consecutive disasters". As such (and also been mentioned in the abstract) I think the formulas that are presented are a generalized formulation, rather than a mathematical framework.

**ANSWER 1**

We agree that the term "mathematical framework" may overstate the level of theoretical innovation. The equations we present are indeed **intentionally simple** and **designed to generalize and integrate concepts from existing literature**, including recovery dynamics and multi-hazard vulnerability.

In the revised manuscript, we will therefore:

- reduce the emphasis on the formal mathematical component,
- clarify that the role of the theoretical formulation is to provide a transparent and implementable backbone for the Python tool, and
- replace the term "mathematical framework" with "Python-based modelling tool" or similar wording, both in the abstract and throughout the manuscript.

This change in terminology and emphasis will better reflect the actual contribution of the paper, which is to make these concepts operational and reproducible in a coherent multi-hazard modelling environment, rather than to introduce a fundamentally new mathematical theory.

**COMMENT 2**

And I think that also links to the lack of novelty of this study. While nicely written up, what is actually new in this formulation? Is the novelty that not too many paper wrote this up in this way? Perhaps. But then I would still expect much more clear examples with a clear time dimension included that shows how this really works. Or is the key element of this work the code behind this study? But would it perhaps not have been better to submit this to a journal like Journal for Open Source Software (JOSS), to emphasize the open source availability of the modelling framework, instead of pushing this into "a new mathematical framework". I do also like this random event

generator in the github repo. Can indeed be nice to play around with stress testing assets in a specific area in a multi-hazard concept. But I don't think I read that really in the paper?

**ANSWER 2**

We acknowledge that the novelty of our work does not lie primarily in the individual equations, but in the way they are combined into a **unified, operational, and time-dependent modelling tool**. In particular, the tool explicitly links:

- multi-hazard interactions
- recovery dynamics
- multi-hazard vulnerability

within a single, modular Python implementation that supports both retrospective and scenario-based analyses.

In the revised manuscript, we will:

- streamline the presentation of the formalism and focus more on how the model behaves over time under different input conditions, explicitly showing how total damage changes as recovery, exposure, and vulnerability parameters are varied;
- include clearer examples and figures that illustrate the time dimension of the model, including the evolution of damage and residual vulnerability between consecutive events:
- present the **two main modes of application** of the tool:
  - forensic use (ex-post analyses based on observed damage and hazard data);
  - anticipatory/ planning-oriented use (scenario simulations to explore possible future outcomes).

Regarding the choice of the journal, we agree that the Python implementation and its open-source nature are central to our contribution. However, we also argue that the paper goes beyond the scope of a pure software presentation article, since it provides a methodological contribution to multi-hazard impact assessment, and a real-world case study that demonstrates how this structure can be used for both validation and scenario analysis. For this reason, we believe NHESS remains an appropriate venue.

We appreciate the reviewer's positive comment about the random event generator included in the repository. In the revised manuscript, we will more clearly describe

this component and briefly discuss its potential use for stress-testing assets in multihazard contexts, as suggested by the reviewer.

**COMMENT 3**

Now, from the Puerto Rico case study example, it is really not clear how actually the recovery response is modelled. It is also highlighted in the discussion that many unknowns are still there with respect to the recovery dynamics. But how are those recovery dynamics actually incorporated in the modelling as presented here. I see it from the mathematical formulations, but not in the application? There does not seem to be a time dimension included, but just an implementation on the fragility curves in a multi-hazard setting?

**ANSWER 3**

We agree that, in the current version of the manuscript, the recovery modelling and its time dynamics in the Puerto Rico application are not sufficiently explained. In the revised manuscript we will:

- provide a clearer description of **how the recovery function Rj(t) is implemented in the Python tool** and how it updates the state of the asset between one event and the next:
- explain that, in principle, the tool allows for a wide range of recovery trajectories (linear, exponential, logistic, etc.), with different durations and shapes depending on the asset type and context;
- clarify that, for Puerto Rico, detailed empirical recovery data were not available, and therefore, instead of assuming a single, known recovery trajectory, we **explored different plausible states of residual damage** at the time of the earthquake by modifying the parameters of the fragility curves (e.g., median and dispersion) to represent varying levels of incomplete recovery.

In other words, the time dimension is incorporated through the sequence of events and the evolution of the asset state between them, which in the Puerto Rico case is represented by a set of alternative residual damage scenarios rather than by a fully data-driven recovery curve.

**COMMENT 4**

The code that is presented behind this paper looks nice, and is cleanly written up. There I can see that (I think) in run\_framework.py that the recovery curve indeed

determines the starting state of the assets when the new hazard hits. However, this is not really clear from the Puerto Rico example.

With reference to the fragility curves, because there is no clear implementation of this recovery aspect in the application, there does not seem to be much novelty in the fragility curves. It is almost a directly implementation from HAZUS?

**ANSWER 4**

In the revised manuscript, we will:

- explicitly state that, in the computational implementation, the recovery function (or the assumed residual damage scenario) determines the initial state of the asset at the onset of the second hazard, as the reviewer correctly inferred from the code;
- add a short subsection or paragraph in the methods/application section explicitly linking the key code routines to the conceptual steps (damage from the first event
  → residual state → updated fragility for the second event).

Regarding the fragility curves, we confirm that the baseline fragility curves are indeed derived from HAZUS for earthquake damage, but for the Puerto Rico case, we systematically modify these curves to represent different levels of residual damage induced by the hurricane (e.g., shifting median values and, where appropriate, changing dispersion), thus **creating state-dependent fragility curves that reflect incomplete recovery**.

We will clarify that the novelty does not lie in proposing entirely new empirical fragility models, but in how standard fragility functions are dynamically adapted and integrated within a consecutive multi-hazard framework, implemented in an open and reusable way. This is precisely the type of operational integration that the tool is designed to support.

**COMMENT 5**

And to continue on this crucial point of the time component, here those non-physical asset damages are really becoming a key element. Dynamic modelling of the recovery process (which is a key element of the mathematical framework as presented here) goes really beyond the physical asset damages. And specially beyond what the example now shows with the "simple" multi-hazard or the modification of the fragility curves with different state dependencies.

I think many of my points are also summed in section 4.1 and section 4.2. For example, I would expect that a paper with this title would show mathematical formulations that move away from this simplification. This also links to the key points mentioned in the "Better understanding and modelling recovery dynamics" and "Evaluation dynamic exposure over time". I understand that the equations as presented in this paper could perhaps provide a starting point to all of this, but I feel like some of these elements should be included already to warrant publication on a leading journal in the field, such as NHESS.

**ANSWER 5**

We fully agree that a comprehensive dynamic modelling of recovery should eventually encompass not only direct physical damage, but also non-physical dimensions such as functionality, indirect economic losses, and socio-economic recovery trajectories. We also agree that this broader scope represents an important direction for future research.

At the same time, we would like to clarify that **the present tool is explicitly focused on direct physical damage to assets**, and uses physical damage (and associated loss) as the primary metric to describe the effect of consecutive and concurrent hazards over time.

In the revised manuscript, we will:

- more clearly delimit this scope in the Introduction and Discussion, stating that indirect impacts and non-physical dimensions of recovery are outside the current implementation and are treated as future developments;
- strengthen Sections 4.1 and 4.2 by explicitly linking them to the reviewer's remark, emphasizing that detailed, empirically based recovery models, dynamic exposure models, and functionality-based impact metrics are essential next steps to build on the foundations provided by our current tool.

We will also underline that we intend to offer a starting point, a flexible, physically grounded tool that can later be extended to non-physical damages and more complex recovery representations, as more detailed data and models become available.

**COMMENT 6**

So to conclude: I think the paper is published with the idea to publish the python code. I do not think that the theory presented in this paper (and the case study

results) are very exciting by themselves, and are not necessarily very novel. However, the code that is presented contains a bunch of nice elements, that I have not really seen in, for example, the DamageScanner or Delft-FIAT. There might be elements of this code in CLIMADA, but that one is sometimes a bit hard to understand what's all in there. As such, I propose to rather focus on writing an article, to a suitable journal, that mostly focuses on the publication of the code.

**ANSWER 6**

We sincerely appreciate the reviewer's positive assessment of the code and the recognition that it contains useful elements not readily available in other tools. We also understand the concern that the paper might be perceived as primarily motivated by the desire to publish the code.

Our intention, however, is to combine:

- a generalized, time-dependent conceptual structure for multi-hazard physical damage;
- a transparent and modular Python implementation;
- a **real-world case study** that demonstrates how the tool can be used for both forensic validation and scenario-based exploration.

In the revised manuscript, we will:

- shift some of the lower-level implementation details to supplementary material or code documentation;
- keep the main text focused on **methodological insights** arising from the simulations (e.g., underestimation of damage when residual damage is neglected, sensitivity of results to assumed residual states);
- clearly articulate how the tool advances multi-hazard impact assessment practice, rather than being solely a software release.

We agree that, in principle, a separate software-focused article (e.g., in JOSS or GMD) could complement this work. However, we believe that the combination of conceptual formulation, operational implementation, and case-study-based methodological discussion makes this manuscript suitable for NHESS, whose scope includes methodological advances in natural hazard and risk assessment supported by numerical tools.

Finally, we will explicitly highlight the **open-source and modular structure** of the code as a key asset for scientific replicability and extensibility to other hazards, asset

types, and regions. This, we hope, will underline that the code is not an accessory component, but an integral part of a broader methodological contribution.